

# A study of repetitive sequences in the genome of *Sinopodisma qinlingensis*

Xiongyan Yin, Nan Zhang, Xiaoyu Li, Lijia Gan, Yimeng Nie and Yuan Huang

College of Life Science, Shaanxi Normal University, Xi'an, Shananxi, China

## ABSTRACT

The family Acrididae characterized by a remarkable genome size and a significant proportion of repetitive sequences. In this study, we find a considerable characteristics by examining the *Sinopodisma qinlingensis*, which has an average genome size within the range observed in Acrididae. The genome size of *S. qinlingensis* was determined to be 11.37 pg for females and 10.95 pg for males using flow cytometry. The analysis of low-coverage sequencing data revealed that the total repeat content of the genome was 63.58%, with long terminal repeat (LTR) elements accounting for 17.74% of the genome contents. Phylogenetic analysis of the reverse transcriptase (RT) domains, which are found within LTR and LINE sequences with consistent conserved motifs, showed that LTR elements belong to multipl within a monophyletic branch. This finging suggests that LTR elements did not originate independently, but rather shared a common evolutionary history. Additionally, the content of Ty3-Gypsy sequences within LTR elements was found to be significantly increased. Fluorescence *in situ* hybridisation (FISH) showed that most satellite DNA and LTR elements exhibited an aggregated distribution pattern on the chromosome.

## INTRODUCTION

Repetitive sequences are widely regarded as the primary driving force of genome evolution, with the capacity to influence a range of biological processes including genome rearrangement (*Brosius, 2003*; *Dolezel, Greilhuber & Suda, 2007*), replication (*Bailey, Liu & Eichler, 2003*), and transfer, which can result in the emergence of new functions (*Cabral-de-Mello & Palacios-Gimenez, 2025*; *Palacios-Gimenez et al., 2020*). The integration of repetitive sequences into genomes can potentially disrupt gene function, thereby presenting a significant challenge to genome sequencing and analysis. Consequently, a more profound comprehension of the structural and dynamic characteristics of repetitive sequences is imperative for genome-related research (*Liu, 2007*). Whole genome sequencing has demonstrated a positive correlation between genome size and repeated sequences (*Du et al., 2016*; *Gregory, 2005*, *2002*). Such variations in repetitive sequences are considered a significant factor contributing to the variation in genome size among eukaryotes (*Ruiz-Ruano et al., 2018*), with the relative frequency of their insertions and deletions playing a pivotal role in the evolution of genome size (*Alfsnes, Leinaas & Hessen, 2017*; *Carta & Peruzzi, 2016*; *Dolezalova et al.,*

Corresponding author
Yuan Huang, yuanh@snnu.edu.cn

*2024*). The significance of repetitive sequences in genome evolution has been extensively acknowledged in Orthoptera, a taxon characterised by a substantially enlarged genome among insect taxa (*Zhao et al., 2025*). Nevertheless, the mechanism of continuous genome enlargement during the evolution of Caelifera insects (belonging to Orthoptera) and its relationship, content and distribution pattern of repetitive sequences remain to be elucidated.

The advent of next-generation sequencing technologies has had a significant impact on the study of repetitive DNA, which, due to its rapid evolutionary rate, has become an important tool for the study of genome evolution (*Li et al., 2023*). This has enabled the tracing of common ancestry across species and facilitated chromosome identification and differential analyses (*Huber, Voith von Voithenberg & Kaigala, 2018*). Consequently, highly complex repetitive DNA has been successfully employed in the study of insect taxa. In Orthoptera insects, studies of repetitive DNA have revealed unique organisational and evolutionary characteristics (*Camacho et al., 2015*; *Ferretti et al., 2020*; *Yuan et al., 2021*). Satellite DNA is a class of DNA that exhibits a specific distribution on chromosomes, consisting of short repetitive units (*Majid et al., 2024*; *Zhongying et al., 2020*). The distribution of satellite DNA on chromosomes can be localised using various techniques, among which FISH is one of the commonly used methods. The FISH technique employs the use of labelled probes, which hybridise with specific DNA sequences on a chromosome, thereby facilitating the determination of the distribution of satellite DNA on the chromosome (*Navarro-Dominguez et al., 2023*; *Ruiz-Ruano et al., 2015*). Orthoptera is the only known group in the insect class with significant genome expansion. A recent study of 59 insects in the family Acrididae demonstrated that the genome size ranged from 6.60 pg to 19.35 pg, with repetitive sequences accounting for 83.58% of the gigantic genome. The *Sinopodisma* has experienced genome gigantism, with an average genome size exceeding 11 pg. The genus *Sinopodisma* is endemic to China, with its distribution restricted to the country. This genus is characterized by small body size and degenerated wings, and it predominantly inhabits mountainous regions at elevations exceeding 850 m (*Zhongying et al., 2020*). The large genome and unique morphology of *Sinopodisma* make the analysis of its genomic "dark matter" a key focus for researchers.

The Acrididae family has been observed to exhibit gigantism in genome size, with an average genome size that exceeds 11.91 Gb (*Li et al., 2022*; *Qiu et al., 2024a*; *Sun et al., 2023*; *Ye, Shi & Yin, 2017*), the genome of *S. qinlingensis* is significantly larger than that of *Locusta migratoria manilensis*, which has been previously described. This suggests that *S. qinlingensis* may be considered a medium to large Orthoptera (*Huang et al., 2013*; *Verlinden et al., 2020*). Consequently, this article focuses on *S. qinlingensis* (*Huang et al., 2013*; *Zheng, 1996*) as the research subject, employing genetic techniques such as genome exploration sequencing and flow cytometry (*Yuan et al., 2021*), and fluorescence *in situ* hybridization (FISH) to investigate the genome size (*Tao et al., 2023*; *Chen & Song, 2023*), repeat sequence types, content, and chromosome localization (*Dolezel et al., 2003*) of *S. qinlingensis*. This investigation provides valuable resources for the study of genome size and repeat sequences in Orthoptera insects.

In order to better understand the content of repeats and their molecular distribution characteristics in the evolutionary process of *S. qinlingensis* and verify the role of repeats in genome size variation, fluorescent hybridization technology is used to test the evolutionary hypothesis, thereby reflecting its evolutionary mechanism and pathway. The application of these research methods provides comprehensive and in-depth insights for the genome study of *S. qinlingensis*, and offers a new perspective for the study of its evolution and genetic diversity. It also provides valuable data and references for the genomics study of medium and large insects in the Acrididae family.

## MATERIALS AND METHODS

### Sample collection, genome size, and next-generation sequencing

The research subject *S. qinlingensis* was collected from Xunyangba Township, Ningshan County, Ankang City, Shaanxi Province, China (Longitude 108°33′0.310″E, Latitude 33°32′56.778″N). Live samples were subjected to genome size estimation using flow cytometry (FCM). The rest of the samples were treated with liquid nitrogen and stored in a −80 °C refrigerator for sequencing and subsequent experiments.

Genome size was measured using flow cytometry (*Hen et al., 2021*; *Robinson et al., 2023*), with the reference standard sample being male *L. migratoria manilensis* (*Verlinden et al., 2020*). The genome size of *S. qinlingensis* was calculated using the following formula:

$$X = \frac{AX}{AC} \times Cpg$$

where:

X represents the nuclear DNA content of *S. qinlingensis* (in pg);

AX represents the fluorescence intensity of *S. qinlingensis*,

AC represents the fluorescence intensity of male *L. migratoria manilensis*;

C is the nuclear genome size of male *L. migratoria manilensis* which is 6.20pg.

The sequencing process employed Illumina sequencing technology (*Blair & Durrance, 2024*; *Haendiges et al., 2021*; *Stoeck et al., 2024*) to construct a 350 bp insertion fragment library and performed sequencing with paired-end 150 bp reads (PE-150).

### Analysis of repetitive sequence and construction of phylogenetic tree of repetitive elements

We used FastQC software to perform checks on the 150 bp paired-end read data generated by whole-genome sequencing (WGS). Random sampling was conducted using the SeqTK tool, repeated three times, and three selected samples of *S. qinlingensis* second-generation data (0.1X) were analyzed for total repetitive sequence content and type. Subsequently, we then used the RepeatExplorer_Utilities tool to merge the paired-end data to meet the requirements of the RepeatExplorer software (*Haq et al., 2022*). Finally, we analyzed the repetitive sequences using the RepeatExplorer2 software on the Galaxy platform (re-characterized by dnaPipeTE software annotations).

The contigs file from the aforementioned analysis were analysed using DANTE on the Galaxy platform, with the following output parameters specified: min_length = 100,
max_length = 10,000, and similarity_threshold = 0.8. This process yielded the LINE information, and the Genome Browser software was subsequently used to extract the target sequences. MEGA10.0 (Molecular Evolutionary Genetics Analysis) (*Tamura, Stecher & Kumar, 2021*) was then employed to construct separate phylogenetic trees for LTR and LINE repetitive sequences of *S. qinlingensis* using the maximum likelihood (ML) method (*Myung, 2003*). The optimal partitioning model of proteins, recommended by Model Selection (an inbuilt tool of MEGA 10.0), was utilised to construct the phylogenetic tree, which was then refined using Figtree software (*Mohammadi, 2017*).

## Fluorescence *in situ* hybridization localization of repetitive sequence positions in the *S. qinlingensis*

### Preparation of films and probes

A solution of 0.05% colchicine at a volume of 6–8 µL was injected. A total of 6–8 h later, the testes of *S. qinlingensis* grasshoppers were dissected and placed in 0.075 mol/L hypotonic potassium chloride solution for 15 min, followed by fixation, squashing, staining with a 5% Giemsa staining solution, and storage at −20 °C for later use.

The PCR labeling method (*Buchner, 2024*; *Serpieri & Franchi, 2024*; *Weidner et al., 2024*) was employed for amplification to generate the probes. Genomic DNA from *S. qinlingensis* was successfully extracted and served as the template. Biotin-labeled nucleotides (biotin-11-dUTP) were incorporated into the reaction (*Milani et al., 2021*). Details of the PCR reaction system and parameters are provided in the Appendix (Tables S1 and S2). Genomic DNA was extracted from the *S. qinlingensis* genome, and its concentration was *S. qinlingensis* genome DNA was 360.141 µg/ml with an OD value of 1.899, and the OD value was within the requirement of 1.8–2.0. Subsequently, DNA was 360.141 µg/ml with an OD value of 1.899, and the OD value was within the requirement of 1.8–2.0. The genomic DNA was fragmented to a length of approximately 500 bp to meet the requirements for subsequent sequencing. The fragmented DNA underwent end repair, and sequencing adapters were added at each end of the DNA fragments to ensure compatibility with the sequencing platform. The libraries were purified using agarose gel electrophoresis to remove DNA fragments lacking adapters.

Finally, the libraries were quantified using quantitative PCR to ensure that their concentrations met the requirements for sequencing. Primers derived from five satDNAs and five LTR repeats that satisfied the experimental conditions were screened through pre-testing for PCR amplification, and the amplified products were the desired probes. The amplification products were analyzed *via* electrophoresis and exhibited the correct size of the target bands along with clear, single bands. Electropherograms of the amplified products from satDNA PCR are shown in Figs. S1A and S1B. We observed the target bands in the expected locations, which appeared thicker and brighter. The concentration of the probes was measured using an ultra-micro UV-visible spectrophotometer, and the concentration of the 10 probes was found to range of 700–900 µg/ml.

**Table 1 Repetitive sequence types and contents of S. qinlingensis (three samples).**

| Types | Sample 1 | Sample 2 | Sample 3 | Average value | SD |
|---|---|---|---|---|---|
| 45S_rDNA | 0.36% | 0.42% | 0.35% | 0.38% | 0.04 |
| Satellite | 2.62% | 4.24% | 2.05% | 2.97% | 1.14 |
| LTR | 18.72% | 16.64% | 17.87% | 17.74% | 1.05 |
| Penelope | 1.98% | 2.91% | 1.77% | 2.22% | 0.61 |
| LINE | 9.79% | 9.48% | 9.84% | 9.70% | 0.20 |
| Maverick | 9.55% | 17.22% | 7.56% | 11.44% | 5.10 |
| Helitron | 0.56% | 0.21% | 0.23% | 0.33% | 0.20 |
| Plastid | 0.01% | 0.01% | 0.02% | 0.01% | 0.01 |
| Mitochondria | 0.10% | 0.09% | 0.10% | 0.10% | 0.01 |
| Unclassified mobile_element | 7.00% | 0.65% | 7.64% | 5.10% | 3.86 |
| Unclassified repeat | 13.15% | 11.87% | 15.74% | 13.59% | 1.97 |
| Total | 63.84% | 63.74% | 63.17% | 63.58% | 0.36 |

***Fluorescence in situ hybridization verification***

To validate the 10 repetitive sequences, we performed FISH experiments (*Qiu et al., 2024b*). Among them, five satellite DNA probes were labelled with biotin and hybridised to *S. qinlingensis* chromosomes, separately. Finally, we estimated the percentage abundance of repetitive sequences and their copy number in *S. qinlingensis* using the RepeatMasker software (*Tempel, 2012*).

## RESULTS

### Determination of the genome size of *S. qinlingensis*

The genome size of *S. qinlingensis* was measured by means of flow cytometry, with the experiment requiring a coefficient of variation (CV) of less than 5% in order to ensure that the quality control was up to standard. The DNA content of the *S. qinlingensis* samples was calculated using formulas based on the fluorescence intensity data obtained by flow cytometry. This ensures the accuracy and reproducibility of the experimental results (Fig. S2). To ensure the reliability of the experimental results, all experiments were repeated thrice, and the mean value of the results from the three experiments was calculated. This calculation yielded a genome size of 11.3677 pg for females and 10.9455 pg for males, with a difference of 0.4222 pg between the two (Table S3).

### Results of repeated sequences of *S. qinlingensis*

Using RepeatExplorer2 and dnaPipeTE to compare the analysis results, we comprehensively analyzed the type and content of repetitive sequences in the *S. qinlingensis* genome. Table 1 lists the results of RepeatExplorer2 analysis of the three 0.1X second-generation data samples, and Fig. 1 illustrates the type and content of repetitive sequences in the *S. qinlingensis* genome resulting from the final analysis. Figure S3 shows the type and content of repetitive sequences in the genome of *S. qinlingensis* as analyzed by dnaPipeTE.

a.

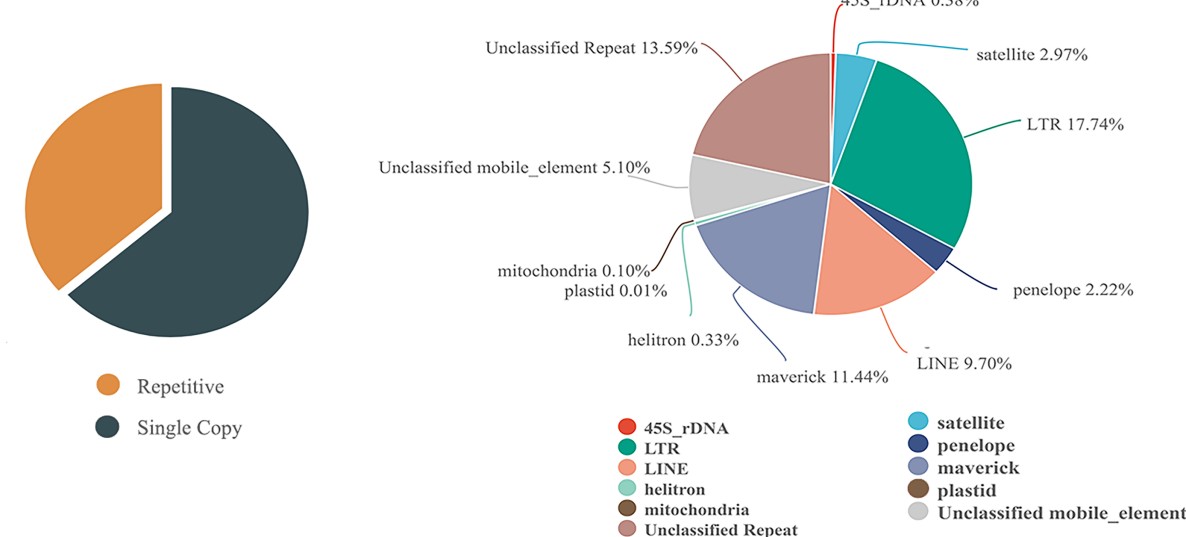

Types of repetitive sequences

Types and proportions of repetitive sequences

b.

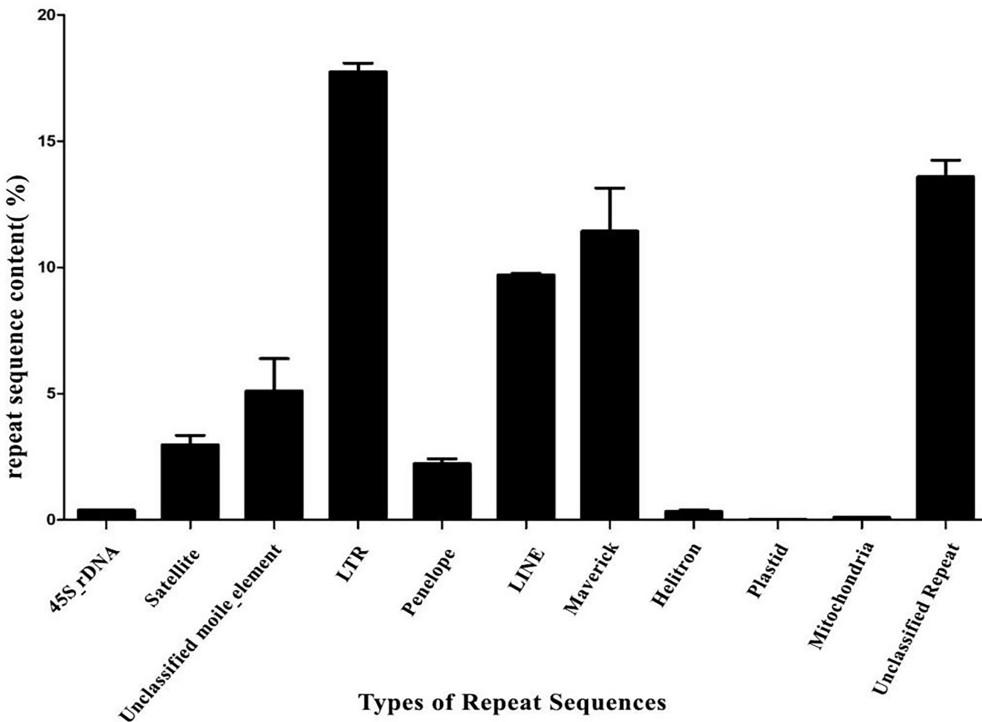

**Figure 1  Repetitive sequence types and contents of *S. qinlingensis*.** (A) Repetitive sequence types and contents of *S. qinlingensis*. (B) Repeat sequence analysis results of *S. qinlingensis* by RepeatExplorer2.

RepeatExplorer2 analysis showed that the total content of repetitive sequences of *S. qinlingensis* was 63.58%. The types and relative contents of repetitive sequences were as follows: 45S rDNA 0.38%, Satellite DNA 2.97%, LTR 17.74%, Penelope 2.22%, LINE 9.70%, Maverick 11.44%, Helitron 0.33%, Plasmid 0.01%, Mitochondrion 0.10%, Unclassified mobile element 5.10%, Unclassified repeat 13.59%. dnaPipeTE software analysis showed that the total content of repetitive sequences of *S. qinlingensis* was 63.65%, which was more consistent with the results of RepeatExplorer2 analysis, LTR 10.99%, LINE 14.12%, SINE 1.59%, DNA 16.55%, Helitron 0.33%, rRNA 0.38%, Low_Complexity 5.10% Simple_repeat 13.59%, This shows that the results of the two software programs are more consistent regarding the total content of repetitive sequences, but the TEs analyzed by dnaPipeTE are obviously not as good as the results of RepeatExplorer2.

The RepeatExplorer2 software (*Novak, Neumann & Macas, 2020*) was utilized to analyse the repetitive sequences of three *S. qinlingensis* samples, The findings demonstrated that the repeat sequence types obtained from the three experiments were consistent, with a total of nine annotated types and some unannotated sequences. The standard deviation of the three samples was calculated, and the results indicated that the content of the majority of repetitive sequences exhibited minimal variation. However, the content of unclassified mobile elements and maverick sequences demonstrated greater variability. The total content of repetitive sequences was found to be similar for the three samples, with only minor differences observed between 63% and 64%.

Although dnaPipeTE has superior performance in terms of runtime and transposon annotation, and is able to complete efficient analyses in a short time, RepeatExplorer2 software shows more obvious advantages when dealing with complex genomes, as it is able not only to automatically annotate transposable elements, but also to concatenate repetitive sequences to provide more intuitive results, RepeatExplorer2 not only automatically annotates transposable elements, but also tandem repeats, providing more intuitive analysis results, making it even more powerful in analyzing repeat composition and distribution patterns.

## Construction of phylogenetic tree for repetitive components

The LTR sequences obtained from RepeatExplorer2 analysis were utilised for the construction of phylogenetic tree (*Zhou et al., 2021*; *Zou et al., 2024*) with the results presented in Fig. 2.

RepeatExplorer2 analyzed a total of 65 LTR repeat sequences, including nine from the Bel-Pao superfamily, 5 Ty1-copia superfamily, 45 Ty3-gypsy superfamily, and six unclassified LTRs. Phylogenetic analysis revealed that the Ty3-gypsy superfamily was the more predominant among the LTR elements and appearing prinarily in clusters, indicating a high degree of homology. This finding is consistent with the evolutionary hypothesis, as LTR elements typically spread and amplify in the genome through reverse transcription and transposition mechanisms reflecting their shared evolutionary histories. Additionally, a small number of sequences from the Ty3-gypsy, Bel-Pao, and Ty1-copia superfamilies, as

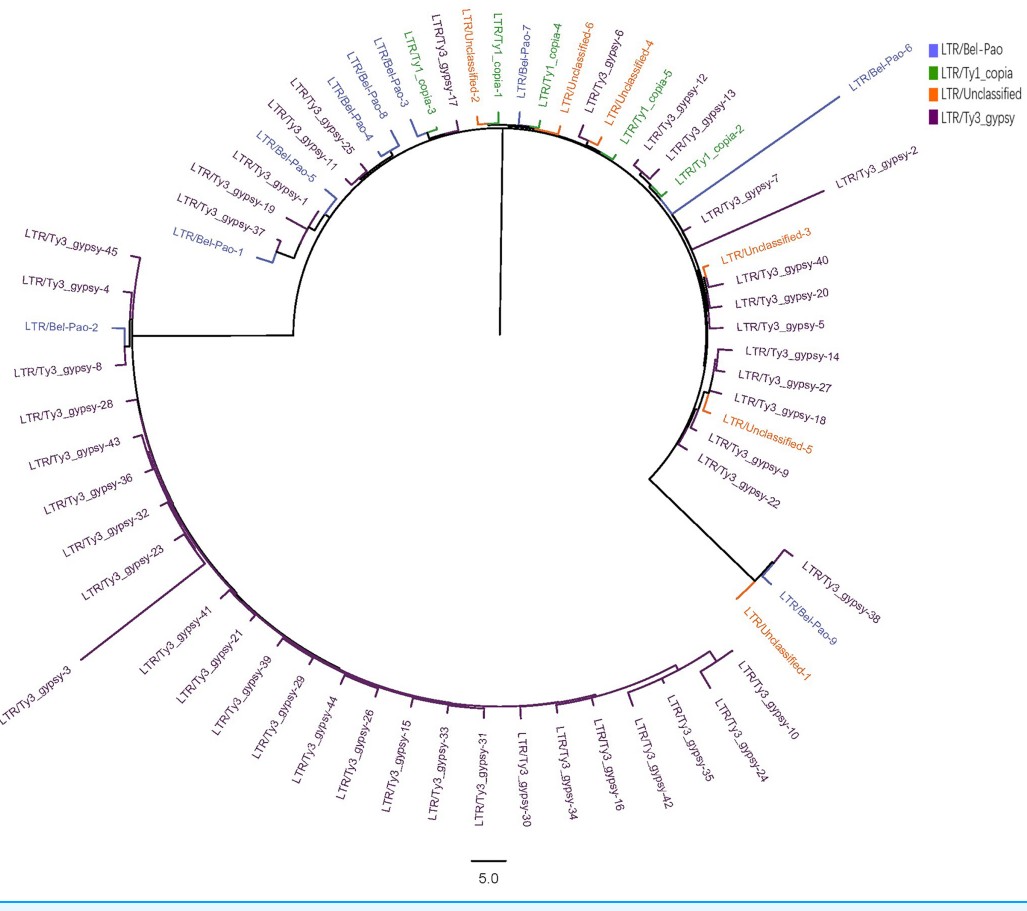

**Figure 2 Phylogenetic tree of LTR in *S. qinlingensis*.**

well as unclassified LTRs, clustered into another group, possibly reflecting genetic exchange or common evolutionary pathways among these superfamilies.

LINE sequences identified through DANTE analysis were subjected to phylogenetic analysis, resulting in the phylogenetic tree shown in Fig. S4. A total of 253 LINE repeats were identified through DANTE analysis. The phylogenetic tree revealed that the majority of LINE sequences exhibited a high degree of homology, suggesting a common ancestral origin. Except for a few LINE sequences (marked in red in Fig. S4), the remaining sequences underwent similar evolutionary changes. This finding suggests that the majority of LINE sequences likely followed analogous evolutionary pathways within the genome, indicating that they may have been subject to similar evolutionary pressures. These observations are consistent with the evolutionary hypothesis, as the homology of LTR and LINE elements reflects their diffusion and amplification processes in the genome, as well as their exposure to similar selective pressures during evolution. This homology not only reveals the mechanisms by which these elements spread within the genome but also suggests that they have undergone complex dynamic changes during genome evolution, influencing genome size, structure, and function.

## Fluorescence *in situ* hybridization

### Karyotyping of S. qinlingensis

We used the spermathecae of the *S. qinlingensis* for chromosome preparation, which were stained with 5% Giemsa's stain and imaged in the microscope to image the resultant figure, and then the autosomes were paired and arranged from largest to smallest as in Fig. 3.

We performed chromosome counts on 10 well-dispersed cells, which showed a chromosome number of $2n\male = 20 + XO$, with an XO type of sex determination mechanism. The grouping of chromosomes was 3L + 6M + S + X. Relative length RL values greater than 10% were large chromosomes, and *S. qinlingensis* had three pairs (L1–L3); relative length RL values within 5–10% were medium-sized chromosomes, and *S. qinlingensis* had six pairs (M4–M9); relative length RL values less than 5% were small chromosomes, and *S. qinlingensis* has one pair (S10); sex chromosome X is a medium-sized chromosome, with a relative length RL value of 8.41%, and the length of sex chromosome X is ranked 6th in the whole chromosome group, see Table 2.

### Fluorescence in situ hybridization verification

In this study, FISH experiments were conducted to validate the aforementioned repetitive sequence elements. Five satellite DNA probes and five biotin-labeled LTR probes were used to hybridize with the chromosomes of the *S. qinlingensis* bungee locust, respectively. The signals were clearly observable under a fluorescence microscope, and the distributions of the five satellite DNA and LTRs (*Xu & Wang, 2007*) on the chromosomes of the *S. qinlingensis* were successfully detected. The relevant parameters the karyotypic analysis were used to plot the karyotypic patterns.

Five satellite DNA distribution patterns were analysed by FISH technique, and the results demonstrated a clustered distribution, but there were significant differences in the number of loci and signal strength:

For satDNA-01, two loci were identified on chromosome L2 at the third position, one locus was detected on chromosome M4 at the proximal midend, and two loci were identified on chromosome M8 at the midend. For satDNA-02, a total of four FISH loci were detected on chromosomes L2, L3, and M6, all exhibiting strong signals. Chromosome L2 exhibited one locus at its terminus; chromosome L3 displayed two loci at the 1/4 and 3/4 positions, respectively; and chromosome M6 possessed a single locus at its extremity. For satDNA-03, a total of 11 FISH loci were detected on chromosomes L3, M5, M6, M7, M8, S10, and X, exhibiting varying signal strengths. The signals on chromosome L3 were weaker, while the remaining chromosomes exhibited stronger signals, with loci predominantly distributed in the terminal or medial regions. For satDNA-04, a total of three FISH loci were detected on chromosomes L1, L3, and M5, all exhibiting weak signals. For satDNA-05, a total of five FISH loci were detected on chromosomes L1, M4, M5, M8, and M9, all exhibiting strong signals. The loci were located at the chromosome extremities, specifically at the one-quarter position or within the intermediate region (Figs. 4 and 5).

The distribution pattern of long terminal repeat sequences (LTRs) on chromosomes is consistent with the of satellite DNA, exhibiting an aggregated distribution. For LTR-01, A total of four FISH sites were detected on chromosomes L3 and M4. For LTR-02, two FISH

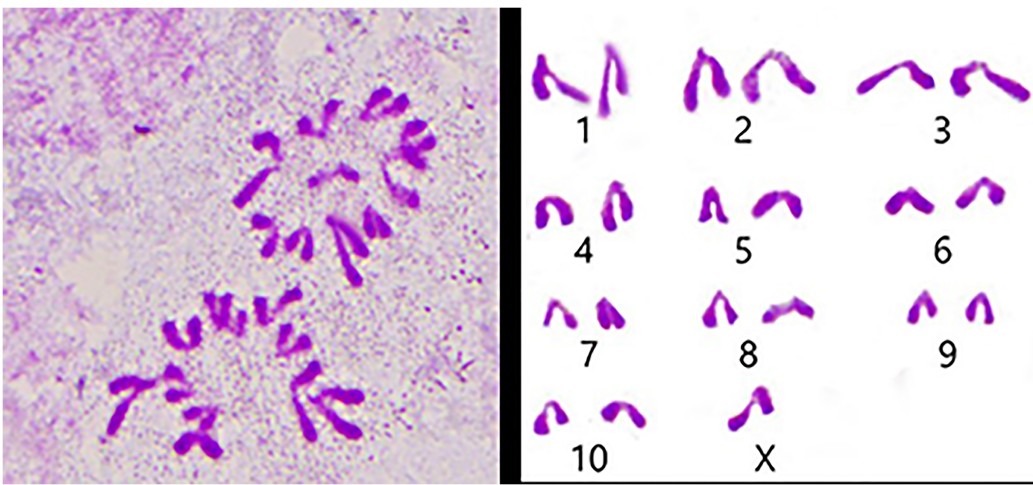

**Figure 3 Karyotype of *S. qinlingensis*.**

**Table 2 Statistic of karyotypic data of *S. qinlingensis*.**

| Chromosome number | Group | Chromosome relative length (%) |
|---|---|---|
| 1 | $L_1$ | 15.70 ± 0.37 |
| 2 | $L_2$ | 13.83 ± 0.32 |
| 3 | $L_3$ | 11.07 ± 0.19 |
| 4 | $M_4$ | 9.63 ± 0.06 |
| 5 | $M_5$ | 8.99 ±0.12 |
| 6 | $M_6$ | 7.74 ± 0.22 |
| 7 | $M_7$ | 6.99 ± 0.25 |
| 8 | $M_8$ | 6.51 ± 0.17 |
| 9 | $M_9$ | 6.03 ± 0.10 |
| 10 | $S_{10}$ | 4.83 ± 0.36 |
| X | X | 8.41 ± 0.17 |

sites were detected on chromosomes L1 and M6, all exhibiting strong signals. For LTR-03, one FISH site was detected near the end of chromosome L1. For LTR-04, one site was detected at the 1/4 position of chromosome L2 and one site in the middle of chromosome S10. For LTR-05, one site was detected at the end of chromosome L3 and one site in the middle of chromosome M7, all exhibiting weak signals (Figs. 6 and 7).

The results of the chromosomal FISH of *S. qinlingensis* are summarized in Table 3. The results demonstrated that, most of the five satellite DNA and five LTR sequences were localized on autosomes, with only satDNA-03 localized on sex chromosomes.

As illustrated in Table 4, the estimated abundance percentage and copy number of *S. qinlingensis* were determined using RepeatMasker software. In addition, the percentage of A+T for each satellite DNA family is indicated.

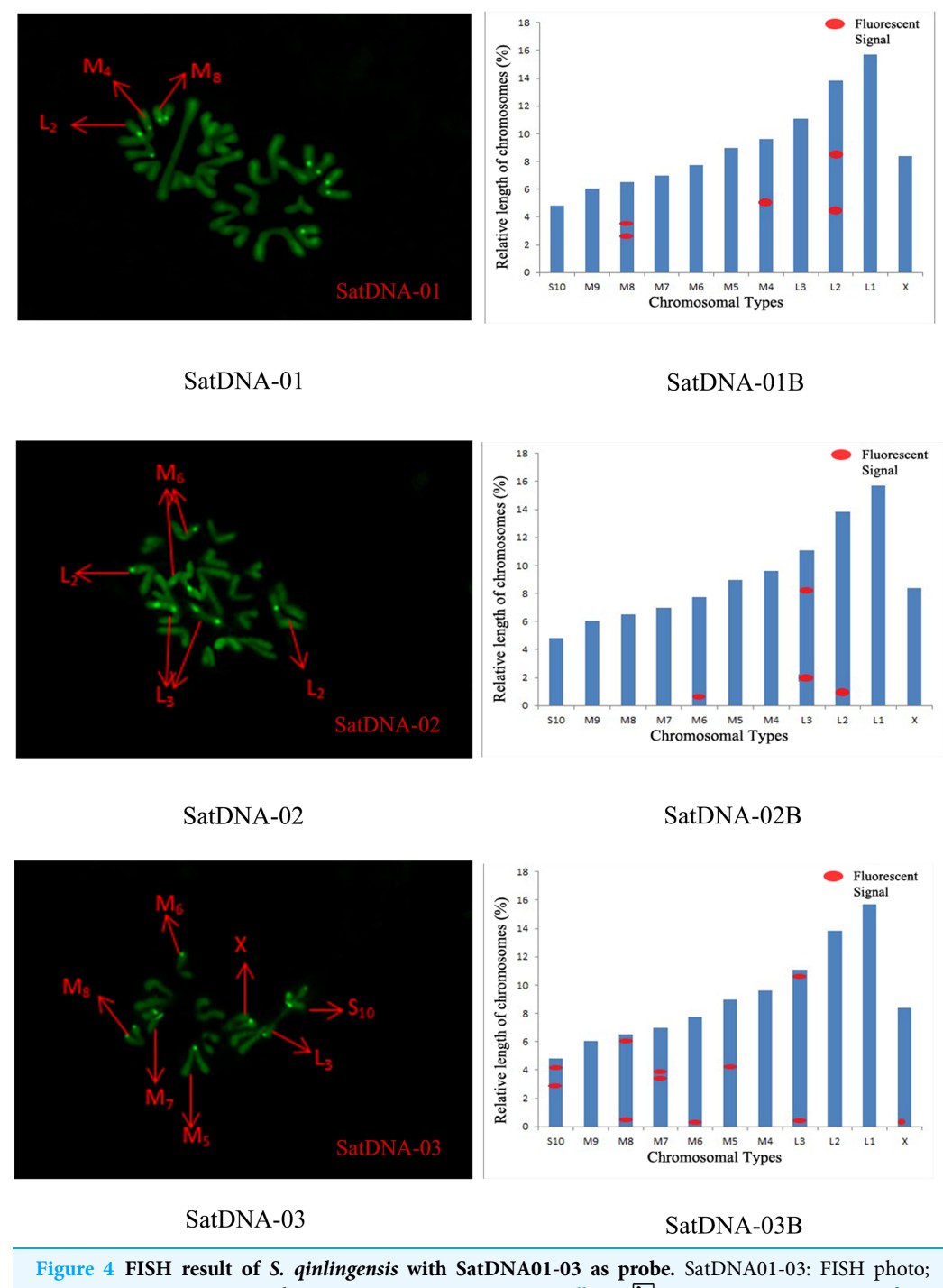

**Figure 4 FISH result of *S. qinlingensis* with SatDNA01-03 as probe.** SatDNA01-03: FISH photo; SatDNA01-03B: Karyotype diagram.

The abundance of the satellite DNA families ranged from 0.015% (Sat-04) to 0.054% (Sat-02), with the total cumulative abundance of the five satellite DNA families in the genome being 0.144%. Among these, Sat-02 (0.054%) and Sat-03 (0.028%) exhibited the highest abundance levels, accounting for more than half of the total satellite DNA content

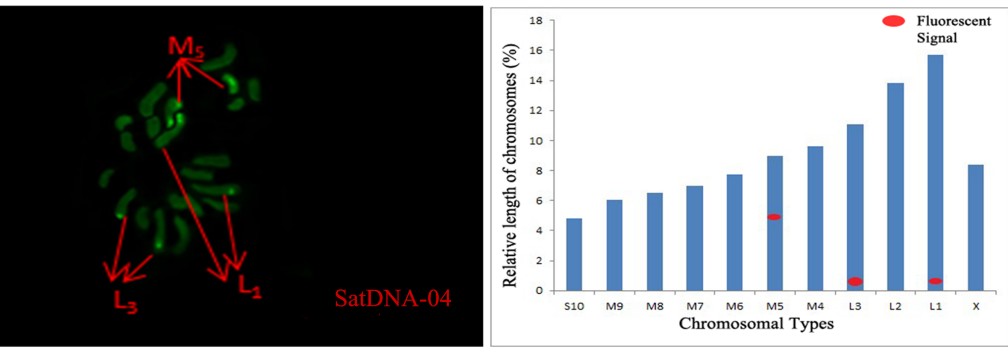

SatDNA-04                      SatDNA-04B

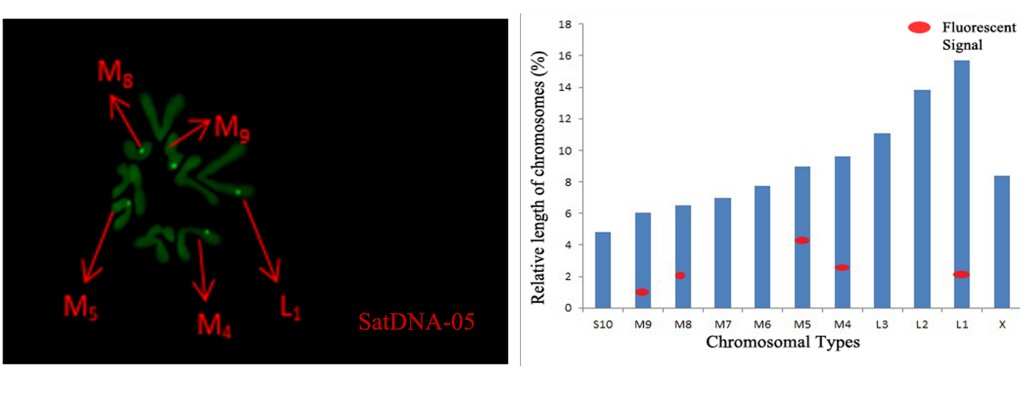

SatDNA-05                      SatDNA-05B

**Figure 5 FISH result of *S. qinlingensis* with SatDNA04-05 as probe.** SatDNA04-05: FISH photo; SatDNA04-05B: Karyotype diagram. 

in the *S. qinlingensis* genome. In contrast, Sat-05 (0.026%) and Sat-01 (0.021%) showed lower abundance levels, while Sat-04 (0.015%) was the least abundant family. Among all five satellite DNA families, Sat-02 and Sat-03 not only showed the highest abundance but also exhibited single-copy numbers of 4,213.22 and 1,0979.18, respectively, significantly higher than those of the other families.

## DISCUSSION

The genomes of Orthoptera insects have demonstrated substantial expansion, with the largest reaching 21.48 Gb. The large genomes of Acrididae insects are primarily attributed to the high content of repetitive sequences. Currently, the highest repetitive sequence content is observed in the *Angaracris rhodopa* genome, which is of a mega-sized genome (16 Gb), where repetitive sequences account for 75.17% of the genome, indicating a significant increase in repetitive sequence content (*Liu et al., 2022*). Compared to *A. rhodopa*, the genome size of *S. qinlingensis* is 11.36 pg, with repetitive sequences constituting 63.58% of the genome. In contrast, the smallest genome among Acrididae insects is that of *L. migratoria manilensis*, with a genome size of 6.60 pg and its repetitive

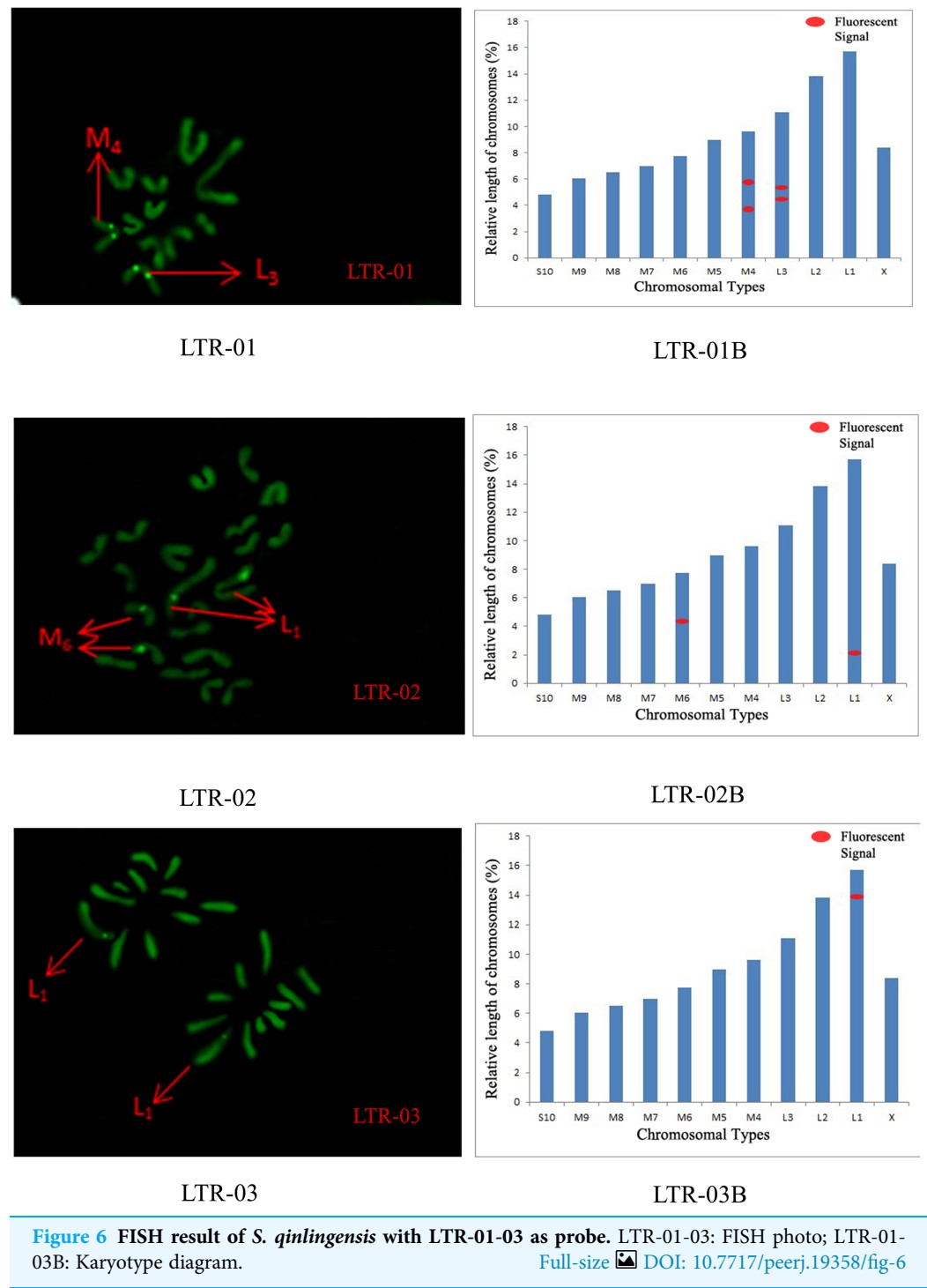

Figure 6 FISH result of *S. qinlingensis* with LTR-01-03 as probe. LTR-01-03: FISH photo; LTR-01-03B: Karyotype diagram.

sequences accounting for 56.83% of the genome (*Cong et al., 2022*). This finding suggests that, compared to the two Acrididae species mentioned, *S. qinlingensis* represents a medium-sized insects within Acrididae. Furthermore, these studies indicate that the proportion of repetitive sequences in the genome continues to increase during the process

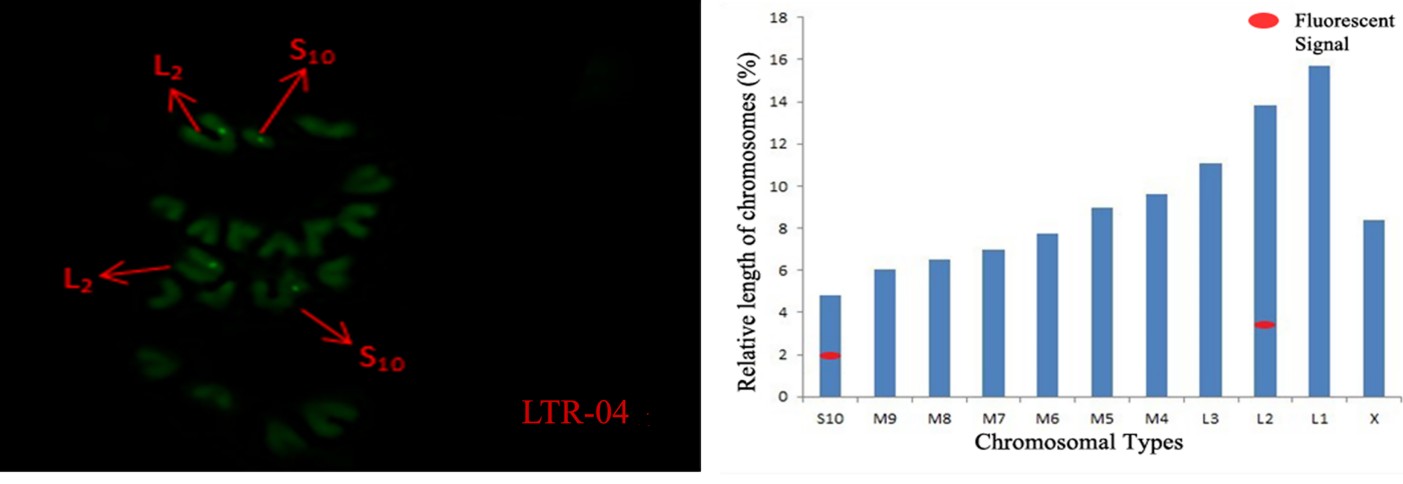

LTR-04                                  LTR-04B

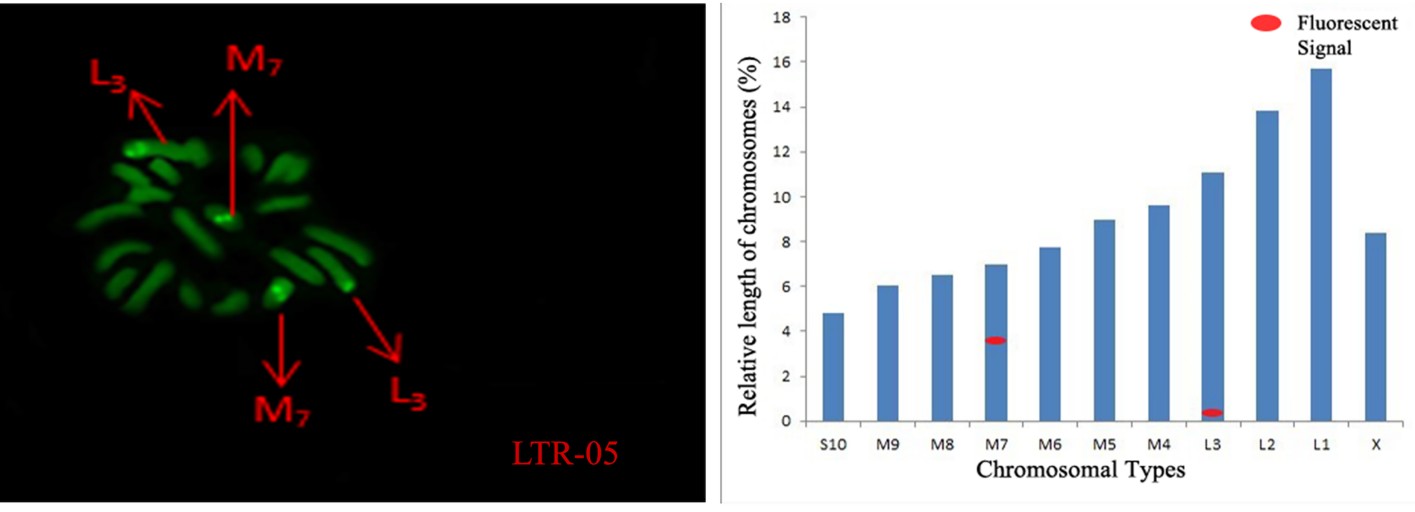

LTR-05                                  LTR-05B

**Figure 7 FISH result of *S. qinlingensis* with LTR-04-05 as probe.** LTR-04-05: FISH photo; LTR-04-05B: Karyotype diagram.

of genome expansion. The expansion of transposable elements (TEs) is considered as potential cause of genome gigantism in Acrididae insects (*Ferretti et al., 2020*). In *S. qinlingensis*, TEs constitute 41.43% of the genome, whereas in the smaller genome of *L. migratoria manilensis*, TEs account for 49.47% of the genome. This discrepancy may be attributed to a bias in the TE database annotation for *L. migratoria manilensis*, as Repbase incorporates consensus sequences of TEs identified using the *L. migratoria manilensis* genome. It is noteworthy that *S. qinlingensis*, harbors a substantial proportion of

Table 3 The distribution of 5 satellite DNAs and 5 LTRs on chromosomes.

| Repetitive sequence | L$_1$ | L$_2$ | L$_3$ | M$_4$ | M$_5$ | M$_6$ | M$_7$ | M$_8$ | M$_9$ | S$_{10}$ | X |
|---|---|---|---|---|---|---|---|---|---|---|---|
| SatDNA-01 | – | * | – | * | – | – | – | – | * | – | – |
| SatDNA-02 | – | * | * | – | – | * | – | – | – | – | – |
| SatDNA-03 | – | – | * | – | * | * | * | * | – | * | * |
| SatDNA-04 | * | – | * | – | * | – | – | – | – | – | – |
| SatDNA-05 | * | – | – | * | * | – | – | * | * | – | – |
| LTR-01 | – | – | * | * | – | – | – | – | – | – | – |
| LTR-02 | * | – | – | – | – | * | – | – | – | – | – |
| LTR-03 | – | – | * | – | – | – | * | – | – | – | – |

Note:
* Fluorescence signals with repeated sequences on chromosomes.
– Fluorescence signals without repeated sequences on chromosomes.

Table 4 The estimated abundance percentage and copy number of *S. qinlingensis* using RepeatMasker software, as well as the percentage of A+T for each satellite DNA family.

| Satellite name | Monomer length | Avg. A + T % | Avg.% Abundance | Avg. copy_number |
|---|---|---|---|---|
| Sat-01 | 905 | 48.39 | 0.021 | 2,583.96 |
| Sat-02 | 1,372 | 47.59 | 0.054 | 4,213.22 |
| Sat-03 | 273 | 60.81 | 0.028 | 10,979.18 |
| Sat-04 | 925 | 52.65 | 0.015 | 1,735.90 |
| Sat-05 | 1,147 | 47.34 | 0.026 | 2,424.41 |

unclassified repetitive sequences (13.59% of the genome), which may signify the emergence of novel TE sequences or even subfamilies during the process of genome expansion. It is particularly noteworthy how transposable elements (TEs) evolve at a significantly faster rate than other parts of the genome. In fact, even among closely related Acrididae species, shared TE sequences are relatively rare.

In contrast tandem repeat sequences, they tend to remain more conerved over time. Studies of Acrididae insects have shown that in two species, *Oedaleus decorus* and *L. migratoria*, the satDNA libraries are still 61% taxonomically incomplete, or approximately 39% taxonomically complete (1.7% per Ma), after about 23 million years of independent evolution (*Camacho et al., 2022*; *Cong et al., 2022*). Notably, the monomers of the satDNAs in these grasshoppers don't have any conserved functional motifs, which is different to other satDNAs such as human centromeric satDNA. This suggests that the satellite DNA in Acrididae insects may be evolving at an accelerated rate. Research on four types of chromosome in the morabine grasshopper *Vandiemenella viatica* suggests that certain satellite DNA families expand in some types, while others contract or are lost (*Palacios-Gimenez et al., 2020*). This supports the hypothesis that satellite DNA in Acrididae genomes has undergone rapid changes in recent evolutionary times (*Liu et al., 2024*).

In this study, the distribution of different probes on chromosomes was determined through the analysis of FISH results. Furthermore, RepeatExplorer2 analysis was used to

characterize their in *S. qinlingensis*, revealing the types and abundance of these repetitive sequences. A comparative analysis indicated that sample selection had a minimal impact on the overall content of repetitive sequences, with only a small proportion of repetitive sequence types being affected. This finding reinforces the conclusion that random sampling had a negligible influence on the repetitive sequences in this study. Analysis of LTR retrotransposons revealed that the Ty3-gypsy family is the most abundant among LTRs, a finding consistent with earlier studies on the prevalence of the Ty3-gypsy family in insects (*Palacios-Gimenez et al., 2020*). The distribution of Ty3-gypsy on the LTR phylogenetic tree suggests that the Ty3-gypsy, Ty3-gypsy and Bel-Pao superfamilies. However, the Ty3-gypsy superfamily does not form a single clade, indicating internal diversity within the superfamily (*Ferretti et al., 2020*). This divergence may be attributed to variations in the activity of the superfamily at different evolutionary time points, which could also explain the predominance of the Ty3-gypsy superfamily. Compared to *L. migratoria manilensis*, the LTR phylogenetic results for *S. qinlingensis* demonstrate the absence of SINE elements, which may be due to evolutionary selection pressure. The integration of SINEs has the potential to compromise the functionality of genes or regulatory regions, resulting in genomic instability and subsequent purging to maintain genomic stability. This underscores the potential involvement of DNA and LTR sequences in genome evolution.

The results from FISH and RepeatExplorer2 analyses reveal that satDNA is mainly localized on autosomes, with more fluorescent signals observed on chromosomes than on LTRs. However, the SatDNA content detected by RepeatExplorer2 is relatively low, indicating that the number of fluorescent signals does not necessarily correlate with the abundance of repetitive sequences.

The present study investigates the distribution characteristics of repetitive sequences in the genome of *S. qinlingensis*, to elucidate their roles in genome evolution. The results indicate that TEs and satDNA play key roles in genome expansion and chromosome structure evolution. Notably the high content and diversity of TEs may be an important factor leading to genome gigantism in Acrididae insects. Furthermore, the enrichment of satDNAs in chromosomal mitotic regions supports their potential role in genome stability and function, thus providing significant insights into the evolutionary history of large genomes in the family Acrididae. Concurrently, this study offers valuable data for understanding genomic variation and evolution in Acrididae insects. These findings provide novel insights into the genome evolution of Orthoptera Caelifera insects and establish a crucial foundation for future studies of genome size variation and repetitive sequence function.

## CONCLUSIONS

The genome size of *S. qinlingensis* was determined by flow cytometry (*Yuan et al., 2021*) as 11.3677 pg in females and 10.9455 pg in males, with a difference of 0.4222 pg. The relatively large genome size is consistent with the genomic characteristics of the Acrididae family, which is known for its large genomes. The total content of repeats was 63.58% in three 0.1X *S. qinlingensis* genome data samples analyzed using RepeatExplorer2, of which

the LTR of LTR retrotransposons accounted for 17.74% of the genome. This is a common feature of the genome of Acrididae and other Orthoptera insects, and these sequences play an important role in genome evolution and structure (*Nie et al., 2024*). Phylogenetic analysis of LTR elements has shown that these elements belong to multiple families within a monophyletic clade, suggesting that these elements share a common evolutionary history and are not of independent origin. In particular, the significant increase in the Ty3-Gypsy sequence in the LTR element highlights a specific trend in the evolution of this genome. These studies imply that the rapid evolution and diversification of transposable elements in the genome of Orthoptera insects can drive genome expansion and increase genetic diversity.

Despite the prevalence of TEs in the genome, certain TE signals were not detected in FISH experiments. This phenomenon may be attributed to various factors, including the dispersion of TEs and the variability of probes, which result in weak or discontinuous signals on chromosomes, that are challenging to detect in FISH experiments. Furthermore, the high diversity of TEs, the potential inadequacy of probes to fully cover all types, and the rapid rate of evolution of TEs may contribute to the fading of some TE sequences in the genome or their replacement by others, thereby preventing the detection of corresponding signals in FISH experiments.

The chromosome number of *S. qinlingensis* is $2n\male = 20 + XO$, and the X chromosome is of a medium-sized. The distribution of five satellite DNA sequences and three LTR sequences on the chromosome was determined by FISH, and all of their sequences showed a cluster-type distribution. The majority of these sequences were concentrated on autosomes, with only satDNA-03 localised on the sex chromosomes. This distribution pattern is consistent with the general organization of repetitive sequences in the genomes of Orthoptera insects, which are usually concentrated in heterochromatin regions. The aggregation and distribution of repetitive sequences may have an impact on chromosome structure and function, thereby affecting the recombination and regulation of genes. The results of the FISH experiments demonstrated that the number of fluorescence signals from satellite DNA exceeded that of LTR, yet the content was lower than that of LTR. This finding suggests that the number of fluorescence signals does not directly correspond to the sequence content.

## ACKNOWLEDGEMENTS

We would like to express our gratitude to Xuanzeng Liu, Xue Zhang, and Ying Mao for their invaluable assistance in the collection and dissection of grasshoppers, as well as their contributions to the FISH experiments.

### Funding

This work was supported by the National Natural Science Foundation of China (grant number 31872217). The funders had no role in study design, data collection and analysis, decision to publish, or preparation of the manuscript.

## Grant Disclosures

The following grant information was disclosed by the authors:
National Natural Science Foundation of China: 31872217.

## Competing Interests

The authors declare that they have no competing interests.

## Author Contributions

- Xiongyan Yin conceived and designed the experiments, analyzed the data, prepared figures and/or tables, authored or reviewed drafts of the article, and approved the final draft.
- Nan Zhang analyzed the data, prepared figures and/or tables, and approved the final draft.
- Xiaoyu Li performed the experiments, prepared figures and/or tables, and approved the final draft.
- Lijia Gan performed the experiments, prepared figures and/or tables, and approved the final draft.
- Yimeng Nie analyzed the data, prepared figures and/or tables, and approved the final draft.
- Yuan Huang conceived and designed the experiments, authored or reviewed drafts of the article, and approved the final draft.

## DNA Deposition

The following information was supplied regarding the deposition of DNA sequences:
The data is available at GenBank: SRR28582605.

## Data Availability

Flow cytometry data is available at Figshare: https://doi.org/10.6084/m9.figshare.27755820.v1.

## Supplemental Information

Supplemental information for this article can be found online at http://dx.doi.org/10.7717/peerj.19358#supplemental-information.

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
