# Peer review of "A study of repetitive sequences in the genome of Sinopodisma qinlingensis"

_PeerJ, doi:10.7717/peerj.19358_

## Round 0.1 · original submission · Major Revisions

While the reviewers felt that the study was of interest, each had issues with either the English language or structuring of the manuscript, as well as issues around analysis.

·

Basic reporting

Literature:
The introduction and background adequately cite relevant works but sometimes refer to redundant references. More development could be carried out in defining the novelty of the research. The authors should highlight the specific contributions of their study to genome evolution and Orthoptera biodiversity.

Figures and Tables:
Figures are relevant but not user-friendly. Readability must improve by increasing font size, optimizing labels, and removing unnecessary overlaps (e.g., Figures 4 and 5). Legends should explicitly describe the data presented.

Language and Format:
The manuscript contains typographical errors and awkward phrasing (e.g., "aggregated pattern," "weak signals were observed"). A thorough proofreading is required to correct grammar, improve phrasing, and ensure clarity.
Standardize all units and terms (e.g., pg for genome size, % for repeat content) consistently across the document.

Introduction:
Expand the section on the ecological or evolutionary importance of Sinopodisma qinlingensis. Why was this species chosen? Is it representative of Orthoptera insects with large genomes? A clearer rationale for the species' relevance to genome evolution studies is needed.

Experimental design

Research Question:
The research question is well defined, but the authors should clarify the broader significance of studying genome repetitive sequences in Sinopodisma qinlingensis. For instance, explain how this study advances the understanding of genome evolution or aids in discerning biodiversity in Orthoptera.

Methodology:

Flow Cytometry:
The description of flow cytometry for genome size estimation is clear, but the formula used for calculating genome size needs to be better formatted for clarity.
Repetitive Sequence Analysis:
The analysis (using RepeatExplorer2, DANTE, and phylogenetic tools) is comprehensive but incomplete. The authors should justify why some sequences remain unclassified and discuss their potential evolutionary importance.
Fluorescence in Situ Hybridization (FISH):
The FISH methods are detailed, but some redundancy exists, particularly regarding probe preparation. Avoid repeating such information unnecessarily.
Library Preparation:
Add a description of the library preparation step to provide complete methodological transparency.

Validity of the findings

Results Section:
Redundancies: Remove repetitive descriptions of results already displayed in tables and figures from the text.
Figures Interpretation: Provide a clearer interpretation of phylogenetic trees in Figures 2 and S2. Discuss how the observed homologies of LTR and LINE elements align with evolutionary hypotheses.
Conclusions:
While the conclusions are supported by the data, they are overly repetitive. Simplify statements regarding the distribution of satellite DNA and LTRs on autosomes.

Evolutionary Implications:
The evolutionary implications of the findings are under-discussed. The authors should address how the observed clustering of repetitive sequences reflects on genome structure and evolution in Orthoptera insects. Comparisons with other Acrididae species would add depth.

Additional comments

The study adopts an excellent methodological approach and addresses an interesting topic in genome evolution. However, major revisions are necessary to improve the manuscript's clarity, interpretation, and readability. Key areas requiring attention include:

Fixing grammar and typographical errors.
Streamlining repetitive content in the methods and results sections.
Improving the presentation and explanation of figures, tables, and data.
Expanding the discussion to cover the broader evolutionary implications of the findings.

Reviewer 2 ·

Basic reporting

In this manuscript, the authors characterize the repetitive DNA landscape of Sinopodisma qinlingensis by a combination of bioinformatics and cytogenetics. The authors used the well established RepeatExplorer2 pipeline to identify TEs and characterize for the first time several satDNA sequences. Also, the authors used flow cytometry to quantify and measure the genome size of S. qinlingensis aiming to correlate the expansion of the repetitive elements and genome size. The data obtained could be of interest to the scientific community of repetitive DNA in insects, however, the manuscript requires major corrections and improvements.
These findings help to expand the knowledge of genome size evolution in Orthoptera species, focusing on the paper of satellite DNA (satDNA) sequences and Transposable Elements (TEs). Overall, the characterization of repetitive elements by Repeat Explorer showed that TEs play a larger role on the composition of S. qinlingensis, supporting the idea that those genomic features are shaping the genome evolution of Acrididae. The authors also confirmed the findings by cytogenetic analysis, reinforcing the idea that satDNAs and TEs are enriched in the pericentromeric areas of the chromosomes.
However, how the manuscript is written does not highlight the data obtained and changes are necessary to improve to clarify the findings. The English language needs to be improved significantly. Throughout the manuscript, several sentences seem to lack coherency, and some terms that are not directly translated to the field terminology. Improving the English language will help the audience to better understand the results and achievements of this study. (E.g. Line 67-69 “The results of fluorescence in situ hybridization indicated that the majority of satellite DNA and LTR elements were clustered on chromosomes”. All DNA sequences are on chromosomes. This sentence is one example of language improvements that are necessary).
The introduction should be expanded regarding the findings of the literature and not focused on the technique, such as it is observed for the cytogenetic tools or next-generation sequencing. The first and second paragraph should be reorganized. The authors initiate the first paragraph introducing the effects of repetitive DNA in eukaryotic genomes but end it with a statement about Caelifera insects (lines 84-86). This last sentence of the paragraph is out of context, mostly because there is not an introduction to insect’s genome evolution prior to that. Also, the second paragraph lacks a broad view of the field of repetitive DNA analyses in Orthoptera and focuses only on the methodological steps used in the manuscript. I suggest the authors use the second paragraph to expose the current knowledge of the field, for example, to describe better what satDNAs and TEs are and how they are organized in Orthoptera and not focus on FISH. The third paragraph needs to be re-written. The paragraph lacks punctuation in general, what leads to the misinterpretation of the content.
The rearrangement in organization should be applied to the Results section. The authors display the table order differently from the presented data. Tables associated with satDNA characterization come after FISH experiments results.

Experimental design

Transposable elements characterization:
The authors need to expand the analysis and discuss why the total number of TEs analyzed is reduced. Only 65 LTR elements were analyzed. If compared to the LINE elements described (253 elements), this overall LTR characterization does not correlate to the genome proportion observed by RepeatExplorer. My major concern is that the sequences analyzed are not fully characterized by the DANTE pipeline. The lack of manual curation or full availability of the RT domains leads to the question is the sequence being analyzed comprises or not a full-length domain. DANTE employs a hierarchical classification system for mobile elements implemented in REXdb. REXdb is mostly focused on plant genomes. Other pipelines, such as RepeatModeler2 (https://github.com/Dfam-consortium/RepeatModeler), are more sensitive and can result in a broader characterization of TEs in this species, such as the one shown in Liu et al. 2024 -Cited in the manuscript- (https://link.springer.com/article/10.1186/s13100-024-00316-x ).
One possibility for the lower number of LTRs elements is the high similarity between those elements, leading to a collapsed assembly of consensus and reducing the total number of copies characterized. It is not clear if the authors used DANTE_LTR to identify the LTR elements. This tool is more sensitive for LTRs and can improve the characterization of those elements. Moreover, to measure the sequence similarity between all TEs characterized I suggest the authors to use a pipeline that compares the sequences in a pairwise manner and show the level of conservation between them like calcDivergenceFromAlign (https://github.com/Dfam-consortium/RepeatMasker/blob/master/util/calcDivergenceFromAlign.pl).
It is also not clear if the TEs analyzed were restricted to one RepeatExplorer analysis, or if all three samples used were used in the identification and characterization of those sequences. Highlighting how the analysis was done will help the readers to understand whether the sequences characterized are the same sequence analyzed twice.
Finally, the authors should make the sequences characterized available, either as a supplementary material or submitted to databases.
SatDNA sequences:
The authors should provide the sequences from the SatDNAs, either as a supplement or as an accession number on NCBI. There is not a clear characterization of those sequences, which also leads to the lack of availability of confirmation of the data. RepeatExplorer2 generates automatically a consensus sequence as an output for TAREAN. Were those sequences the ones used? This also applies to the design of probes. Which sequences were used to design the primers? Therefore, the authors should provide the sequences identified. Also, the authors could use different pipelines to show how variable those satDNA sequences are in S. qinlingensis genome. (e.g., https://github.com/johnssproul/RepeatProfiler)
Moreover, the authors should address the issue of how different samples generate different genome proportions; this is important for the field. By addressing this issue, the authors could help to explain why different experiments may lead to different genomic proportions.
Phylogenetic Analyses:
It is not clear how the phylogenetic analyses were performed. The authors need to clarify the methodology used. The parameters, the model used and how it was achieved, the number of replicates and so on. Did the authors use ML or Bayesian statistics in the phylogenetic reconstruction? Which substitution model was applied? K2P or GTR? Did the authors use the bootstrap method? This method is based on resampling and replications, is used extensively to assess the robustness of phylogenetic inferences.
I suggest the authors clarify and/or include these analyses in the phylogenetic reconstruction section of the manuscript.
FISH: Experiments:
Overall, the FISH probes seem to work, and the data is convincing. I am worried about the over exposition of the probes. On all experiments it is observed a strong background staining, along with the chromosome arms. I understand that this is an output of the secondary staining but could be minimized by adding another round of washes or a more stringent one.
One important point to be addressed either in the results section or discussion is why the TEs signals are restricted to the pericentromeric regions and not anywhere else of the chromosome arms. I do understand that the expansion of TEs is highly associated with new insertions on the pericentromeric regions, however, LTR elements are broadly described to move to euchromatic regions as well. I feel the authors should bring this topic to discussion.
Regarding the presentation of the data. I suggest the authors build a panel with the DAPI signal only, the probe signal only and a merged image. One for the satDNA families, and one for the TEs. That would help the audience, and myself, to understand the distribution of the repetitive DNA in this species. Also, the authors should include a scale bar.
I also recommend the authors to move the description of FISH experiments present at the annexes to the main body of the manuscript. The description present on the annexes has more information than the one present in the main text.

Validity of the findings

These findings help to expand the knowledge of genome size evolution in Orthoptera species, focusing on the paper of satellite DNA (satDNA) sequences and Transposable Elements (TEs). Overall, the characterization of repetitive elements by Repeat Explorer showed that TEs play a larger role on the composition of S. qinlingensis, supporting the idea that those genomic features are shaping the genome evolution of Acrididae. The authors also confirmed the findings by cytogenetic analysis, reinforcing the idea that satDNAs and TEs are enriched in the pericentromeric areas of the chromosomes.
There is no conclusion statement, only one shorter version of the abstract.

Additional comments

Discussion/Conclusions:
This whole section needs to be re-written. Most sentences are just a restatement of the results. The authors should bring the impact of the findings and compare them to the recent results observed in the literature ( e.g., Zhao et al. 2025 https://doi.org/10.1016/j.ygeno.2024.110971 ; Palacios-Gimenez et al. 2020 https://bmcbiol.biomedcentral.com/articles/10.1186/s12915-020-00925-x ; Majid et al. 2024 https://doi.org/10.3390/biom14080915 ).

The first paragraph lacks citations that support the statements. E.g. lines 273; 277; 278; 282. There is only one citation in the final line (289)
The second paragraph of the discussion (lines 290-302) is a good example of what should be done and why the findings in this manuscript are important. Although contradictory, - this paragraph starts claiming that TEs are not conserved and satDNAs are and end up saying the opposite- this paragraph explains and connects the expansion of LTR elements in Acrididae and genome size evolution.
The third paragraph is just an expansion of the Results and does not bring any input into the impact of the FISH results to the field. Line 303-304 “The distribution of repetitive sequences on chromosomes by Fluorescence in Situ Hybridization (FISH) was used to determine the exact location of satellite…”. This sentence should be re-written. Also, Caelifera insects (as a group of interest of the paper) are only cited in the last sentence of Discussion. The last paragraph of the manuscript/Conclusion should bring the impact of the findings and not results from experiments.
Altogether, the authors don’t approach in the discussion the effects of satDNA presence and expansion of TEs in the genome size evolution by comparing the findings of this manuscript with the literature. It is important to highlight the effect that those sequences had in the evolution of S. qinlingensis.

Reviewer 3 ·

Basic reporting

In this manuscript, Yin and colleagues conducted a study on repetitive DNA sequences in the large genome of an Acrididae grasshopper, utilizing both genomic and chromosomal tools. Their findings contribute new insights into the composition and organization of grasshopper genomes. However, despite significant research in this area over recent years, the information remains insufficient to form comprehensive conclusions about genome evolution in this group.
The manuscript’s primary shortcoming is the authors' omission of critical data from prominent researchers in chromosomal and genomic studies of grasshoppers and insects. These include works published in prestigious journals that could provide valuable comparative data and enrich the discussion and introduction. Furthermore, some information in the discussion is either unsupported by citations. Overall, the text requires extensive revision to improve its readability and ensure accurate representation of the field.
Below, I provide a list of works that should be reviewed and incorporated into the manuscript, along with additional suggestions for improvement. By addressing these points, the manuscript can achieve greater clarity, accuracy, and scientific relevance.

Suggested References
Studies on repeats in grasshopper genomes:
1. https://www.nature.com/articles/s41437-020-0327-7#Sec17
2. https://bmcbiol.biomedcentral.com/articles/10.1186/s12915-020-00925-x
3. https://www.mdpi.com/2073-4425/14/2/397
4. https://www.nature.com/articles/s41437-021-00470-5
5. https://link.springer.com/article/10.1007/s00412-017-0644-7
6. https://link.springer.com/article/10.1007/s00412-014-0499-0

The role of repetitive DNAs in insect genomes:
1. https://www.sciencedirect.com/science/article/abs/pii/S2214574524001378

Specific Comments
1. Line 75-79: Include the information about the impact of repetitive elements in insect genomes, as detailed in the referenced in https://www.sciencedirect.com/science/article/abs/pii/S2214574524001378.
2. Lines 87-88: The claim that repetitive sequences have been studied in only a few insect species is misleading. While true given insect diversity, numerous cytogenetic and genomic studies have been conducted in various taxa, including Coleoptera, Hymenoptera, Diptera, Lepidoptera, and Hemiptera. Suggested references:
1. Coleoptera (https://www.mdpi.com/1422-0067/25/17/9214)
2. Hymenoptera (https://link.springer.com/article/10.1007/s00412-021-00764-x)
3. Diptera (https://link.springer.com/article/10.1007/s00412-024-00827-9)
4. Other Lepidoptera (https://link.springer.com/article/10.1007/s00412-022-00781-4)
5. Hemiptera (https://academic.oup.com/biolinnean/article/140/4/518/7258616)

3. Line 114: Abbreviations should be used consistently. For example, after the first mention of fluorescence in situ hybridization (FISH), use "FISH" throughout the text.
4. Methods Section: Include all computational analysis parameters.
5. Line 153: Specify the duration of colchicine treatment.
6. Results Section: Distinguish between methods and results. For example, the statement on line 183 ("male L. migratoria manilensis as a reference standard") belongs in the methods section.
7. Line 265-270: Reorder satellite DNA families based on abundance, as is standard practice.
8. Line 266: Correct the percentage from 0.54% to 0.054%.
9. Line 277 and throughout: Italicize species names consistently.
10. Lines 272-289: Add references to support the discussion of grasshopper genomes. Avoid using unrelated references, such as those on Lepidoptera, unless making a direct comparison.
11. Line 293-295: Reframe the statement about Acrididae insects to avoid generalization based on a single example.
12. Lines 303-319: Rewrite this paragraph with a comparative analysis of previous data to provide deeper insights into repeat organization in grasshopper chromosomes. The suggested references can guide this task.
13. Lines 334-337: Discuss potential reasons for the absence of TE signals, such as their dispersed nature or probe variability, hampering signal detection. Include references to studies that have mapped TEs in grasshopper chromosomes.

Experimental design

'no comment'

Validity of the findings

'no comment'

---

## Round 0.2 · accepted · Accept

Both reviewers were enthusiastic about the revisions made to the previous version of this manuscript, and that you had sufficiently addressed any previous concerns. The manuscript is now ready for publication.

·

Basic reporting

I have reviewed the revised manuscript “A study of repetitive sequences in the genome of Sinopodisma qinlingensis” along with the authors’ responses to the reviewers’ comments. I am pleased to report that the authors have comprehensively addressed the issues raised in my previous review. The improvements in the Introduction, methodology, and Discussion sections have significantly enhanced the clarity and scientific rigour of the study. Additionally, the figures have been revised for better readability, and the supplementary data have been provided as requested.

Although a final round of minor language polishing is advisable, these residual issues do not detract from the overall quality of the work. In light of the substantial improvements, I recommend accepting the manuscript for publication with only minor final editorial adjustments.

Experimental design

No comment

Validity of the findings

No comment

Additional comments

No comment

Reviewer 2 ·

Basic reporting

In this manuscript, the authors characterize the repetitive DNA landscape of Sinopodisma qinlingensis by a combination of bioinformatics and cytogenetics. The authors used the well established RepeatExplorer2 pipeline to identify TEs and characterize for the first time several satDNA sequences. Also, the authors used flow cytometry to quantify and measure the genome size of S. qinlingensis aiming to correlate the expansion of the repetitive elements and genome size.

Experimental design

The authors added new analyses that reinforce the previous data. The analyses are well done and match the current level of knowledge of the field.

Validity of the findings

The authors have improved significantly the organization, language of manuscript, and improved drastically the discussion section.
Overall, I understand that the manuscript meets the criteria for publication.

Additional comments

The authors have addressed my concerns properly, and I understand that the manuscript meets the criteria to be accepted.